# Intelligent Mapping Method for Power Consumption and Delay Optimization Based on Heterogeneous NoC Platform

**Juan Fang** * **, Huan Zong, Haoyan Zhao and Huayi Cai**

Faculty of Information Technology, Beijing University of Technology, Beijing 100124, China
* Correspondence: fangjuan@bjut.edu.cn

**Abstract:** As integrated circuit processes become more advanced, feature sizes become smaller and smaller, and more and more processing cores and memory components are integrated on a single chip. However, the traditional bus-based System-on-Chip (SoC) communication is inefficient, has poor scalability, and cannot handle the communication tasks between the processing cores well. Network-on-chip (NoC) has become an important development direction in this field by virtue of its efficient transmission and scalability of data between multiple cores. The mapping problem is a hot spot in NoC's research field, and its mapping results will directly affect the power consumption, latency, and other properties of the chip. The mapping problem is a NP-hard problem, so how to effectively obtain low-power and low-latency mapping schemes becomes a research difficulty. Aiming at this problem, this paper proposes a two-populations-with-enhanced-initial-population based on genetic algorithm (TI_GA) task mapping algorithm based on an improved genetic algorithm from the two indexes of power consumption and delay. The quality of the initial individual is optimized in the process of constructing the population, so the quality of initial population is improved. In addition, a two-population genetic mechanism is added during the iterative process of the algorithm. The experimental results show that TI_GA is very effective for optimizing network power consumption and delay of heterogeneous multi-core.

**Keywords:** network-on-chip; mapping algorithms; power consumption; delay; genetic algorithms

## 1. Introduction

When dealing with the application requirements of large data size and high real-time requirements, the system platform of multi-core processors working together shows great advantages compared with the single core processor system platform, but it also poses a great challenge to the on-chip system communication capabilities. The traditional System-on-Chip (SoC) communication usually uses the bus structure [1]. The traditional network communication method is adopted to solve the continuous growth of communication demand between the cores on multi-core chip and to implement a better flow control mechanism [2] to cope with the very large scale. A new method to the design of integrated circuits due to factors such as size and communication requirements is needed [3–5].

The design of network-on-chip (NoC) usually contains four aspects [6]: the mapping belongs to the category of modeling and optimization in which the application is applied. The NoC processing element mapping is an NP-hard problem [7]. Therefore, a serious challenge for NoC designers is how to create a preferable algorithm to achieve mapping solutions with low-power consumption and low-latency [8].

Recently, several NoC mapping algorithms for optimizing energy consumption or time delay have been proposed. A multi-application fault-tolerant mapping algorithm was proposed by Khalili et al. [9],

which maps the application kernel map to the idle error-free core by using an heuristic algorithm, and then places other idle kernels on other idle error-free kernels. This algorithm is more functional and energy efficient than previous solutions, but it will cause local crowding around the faulty node and forming hotspot congestion. Murali et al. [10] proposed a method to ensure quality of service (QoS) in mapping physical planning process by satisfying real-time restrain and delay restrain of traffic flow. Compared to traditional solutions, this design can save up to five times the bandwidths. To minimize the power consumption of different benchmarks, Tosun et al [11] suggested a new ILP formula for mapping applications to grid-based NoCs, but this algorithm still has room for improvement, as the consumptions of run time become enormous when the tasks increase accordingly. Macron et al. [12] compares some NoC mapping algorithms for obtaining a low energy consumption communication weighted model. Although the NoC mapping schemes finally achieved by these algorithms have lower power consumption or higher performance, these papers discuss single optimization targets of NoC, and are not applicable to the multi-target mapping problem of NoC heterogeneous processing units.

The contribution of this paper is three-fold. First, we propose a two-populations-with-enhanced-initial-population based on genetic algorithm (TI_GA) mapping scheme for heterogeneous multi-core NoC. The results of the experiment show that the scheme is better than the existing mapping scheme, reducing power consumption and delay. Secondly, we propose to use the method of optimizing initial population and dual population inheritance to compensate for the shortcomings of genetic algorithm (GA) random initialization and often get into the local optimal solution. Finally, the proposed scheme provides a new perspective for heterogeneous multi-objective optimization problems.

This paper is structured as follows: In Section 2, the concept of Two-Step mapping is proposed, and a formal description of the mapping problem for power consumption and delay is made. In Section 3, a mathematical model of power consumption and delay is established, and in Section 4, TI_GA is proposed, and the algorithms introduced in detail. See Section 5 for simulation results and discussion. Finally, we draw a conclusion in Section 6.

## 2. Two-Step of NoC Mapping

In more complex applications, there are many cooperative tasks, which are generally different in nature. Tasks are usually assigned to heterogeneous IP cores that are good at running such tasks. Compared with isomorphic systems, heterogeneous systems can achieve better system performance and lower power consumption. This paper divides the problem of heterogeneous multi-core NoC mapping into two steps: the first step, as shown in Figure 1, assigns tasks in the application task graph to the IP core in a specific strategy according to the task characteristics. The second step, as shown in Figure 2, assigns IP cores associated with tasks in the first step to NoC network resource nodes with some mapping strategy. In these two steps, different allocation and mapping strategies have different effects on delay and energy consumption. In task graphs with a large number of tasks, factors such as area cost are usually taken into account, and multiple tasks are often assigned to an IP core for execution.

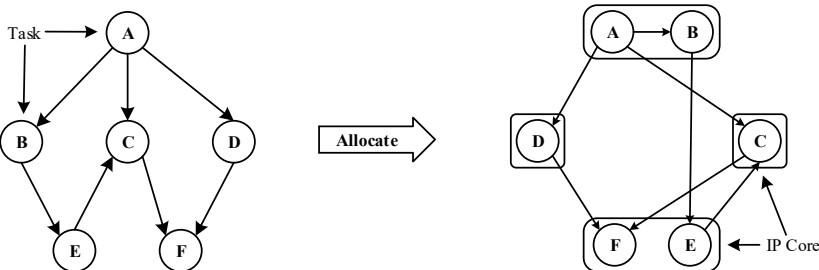

**Figure 1.** The first step of Network-on-Chip (NoC) mapping: task assignment.

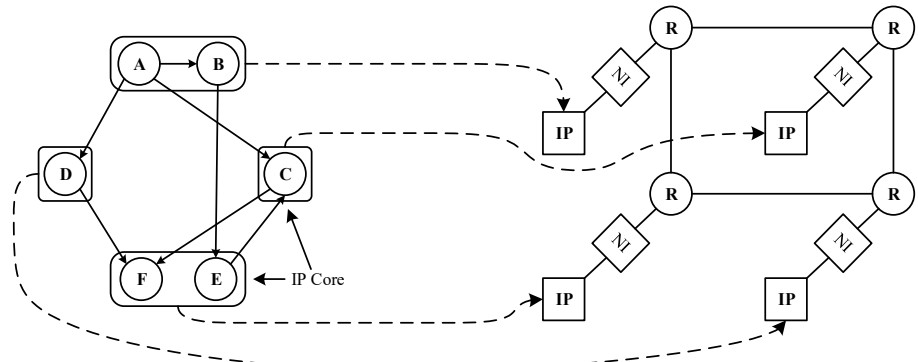

**Figure 2.** The second step of NoC mapping: IP core mapping.

The NoC platform used in the mapping algorithm research in this paper is a two-dimensional mesh (2D-Mesh) to realize the connection between nodes. As shown in Figure 2 (right), its routing node (R) and resource node only correspond to each other. Each routing node and resource node transmit data through network interface (NI). Since the study of the mapping algorithm involves the length of the packet transmission path, in the selection of the routing algorithm [13,14], the NoC platform is specified to utilize the static shortest path routing algorithm, which is easy to address and the conclusion is easily extended to other complex platform architecture.

The chart above shows the process of mapping tasks to NoC platform resource nodes. The following is the basic concepts of this problem.

**Definition 1.** *Given the task communication graph $TCG(T, E)$ is a weighted graph, where $t_i \in T$ denotes the subtasks in the application and $e_{i,j} \in E$ denotes the traffic between $t_i$ and $t_j$.*

**Definition 2.** *Topology diagram $TG(P, L)$, where $p_i \in P$ denotes a resource node in the topology, usually an IP core, and $l_{i,j} \in L$ denotes the communication link between $p_i$ and $p_j$.*

Based on the above definition, the application mapping algorithm for low power consumption and low latency can be described as: given an $TCG(T, E)$ and an $TG(P, L)$, a mapping function $map(T \rightarrow P)$ is found to minimize the overall power consumption and delay.

## 3. Establishment of Mapping Evaluation

This paper will give a calculation model of NoC system energy consumption and delay without considering the number of task nodes and the number of links, so as to evaluate the quality of NoC mapping scheme from the index of energy consumption and delay.

### 3.1. Power Model

The energy consumed in NoC system can be divided into two parts, one is the computing power consumed in the process of computing and processing IP cores in NoC network, the other is the communication power consumed in data transmission in network links. We can express the total energy consumption of NoC system as following formula:

$$E_{NoC} = \sum_i^{IP} E_i^{IP} + \sum_j^{path} E_j^{path}. \tag{1}$$

For data processing of the same size, the energy consumed in IP processing is only related to the type of IP core. The power consumption of processing $\Delta$ bit size data on an IP core can be expressed as:

$$E_\Delta^{IP} = \rho_{IP} \times \Delta, \tag{2}$$

where $\rho_{IP}$ is a constant related to IP core type.

The power consumption in a NoC platform network can be divided into two parts: the router power consumption in NoC and the link energy consumption between NoC routing nodes. At present, many NoC communication power models have been proposed. Here we adopt the commonly accepted communication power model [15]. The specific formula can be described as follows:

$$E_{bit} = E_{Sbit} + E_{Bbit} + E_{Wbit} + E_{Lbit} \tag{3}$$

where, $E_{bit}$ denotes the power consumption of transmitting unit data from one routing node to another, $E_{Sbit}$ denotes the power consumption of the cross-switch, $E_{Bbit}$ denotes the power consumption of unit data in the buffer area of the routing node, $E_{Wbit}$ denotes the power consumption of internal lines, and $E_{Lbit}$ denotes the power consumption of transmitting unit data in the communication interconnection link of the NoC network. Among them, the values of $E_{Sbit}$, $E_{Bbit}$, and $E_{Wbit}$ will not change with the communication environment and conditions in the network, because their values are only related to the internal design of the router, so they can be regarded as an approximate constant. Therefore, we use $E_R$ to express them uniformly, so the above formula can be described as:

$$E_{bit}^{i,j} = n_{i,j} \times E_R + \left(n_{i,j} - 1\right) \times E_{Lbit} \tag{4}$$

where, $E_{bit}^{i,j}$ denotes the power consumption of transmitting unit data from node $T_i$ to node $T_j$, and $n_{i,j}$ denotes the number of routing nodes that pass through in the process of transmitting unit data, that is, the Manhattan distance. Therefore, the power consumption of communication between node $T_i$ and node $T_j$ can be calculated by Formula (5).

$$\begin{aligned} E_{i,j} &= v_{i,j} \times E_{bit}^{i,j} \\ &= v_{i,j} \times n_{i,j} \times E_R + v_{i,j} \times \left(n_{i,j} - 1\right) \times E_{Lbit} \end{aligned} \tag{5}$$

where $v_{i,j}$ represents the traffic between node $T_i$ and node $T_j$.

*3.2. Delay Model*

The delay of the NoC platform includes the time consumed by data processed by the IP core and the time consumed by data transmission between routing nodes in the network. The delay $D$ in the NoC system can be expressed as:

$$D = \sum_i D_i^{core} + \sum_j D_j^{path} \tag{6}$$

where $D_i^{core}$ denotes the time spent on task processing on the $i$-th IP core and $D_j^{path}$ denotes the time spent on data transmission over the $j$-th link. The specific type of task, the type of IP core and the amount of data in the process of task processing determine the time spent on data processing by IP core. The delay $D_\Delta^{IP}$ of processing $\Delta$ bit data on a specific IP core can be expressed as:

$$D_\Delta^{IP} = \rho_{cd} \cdot \Delta \tag{7}$$

where $\rho_{cd}$ is a constant related to the IP core type.

The time consumption of data transmission mainly comes from the physical delay on the connection line and the delay of data packet parsing [16].

The data delay from the source node to the destination node consists of three parts: the delay on the source node NI, the delay on the routers in the transmission path, and the delay on the transmission path interconnection line.

If $\Delta$ bit data is transmitted from network node $a(x_a, y_a)$ through path $p_{ab}$ to network node $b(x_b, y_b)$, the network is a 2D-Mesh topology. Under the condition of using shortest path routing, the calculation formula of time $D_\Delta^{path}$ spent in the transmission process is as follows:

$$D_\Delta^{path} = \Delta \cdot [2\rho_{NI} + \rho_L \cdot d_{ab} + \rho_R(d_{ab} + 1)] \tag{8}$$

where, $\rho_{NI}$, $\rho_L$, $\rho_R$ are the constants of NI, interconnection line and router about transmission time, respectively. Formula (8) assumes that the NI at source and destination nodes is of the same type. $d_{ab}$ is the Manhattan distance between nodes $a$ and $b$.

### 3.3. Multi-Objective Mapping Optimization Model

Traditional network-on-chip mapping adopts a single-objective optimization mapping strategy. With the increase of tasks, network scale, application complexity and so on, the goal of mapping is no longer limited to the optimization of communication energy consumption. If the pursuit of power consumption is only one-sided, it may lead to local congestion in the NoC network, which will cause the delay of executing the application to become longer, thus affecting the quality of service.

Therefore, it is impossible to optimize all the indexes by using the method of optimization mapping. It is a multi-objective optimization problem that all the optimization indexes must be compromised. In this paper, two very important performance indicators in many design constraints of NoC are discussed: system power consumption and system delay. The evaluation function design in multi-objective programming mainly includes linear weighted summation method, ideal point method, maximum–minimum method, square weighted method and so on. Linear weighted summation can highlight the optimization direction by setting each objective weight, and its realization is simpler. Therefore, the linear weighted summation method will be used in the implementation of the algorithm in this paper.

Assuming that the mapping scheme $\varphi$ belongs to the set *MAP* containing all mapping schemes, for multi-objective optimization problems with energy consumption and delay:

$$f = \min\{E(\varphi), D(\varphi)\}. \tag{9}$$

The linear weighted summation method needs to determine the constant $\mu_e$ and $\mu_d$ in advance according to the importance of energy consumption and delay, and satisfy $\mu_e \geq 0$, $\mu_d \geq 0$, $\mu_e + \mu_d = 1$. When $\mu_e = 1$, only power optimization is done, $\mu_d = 1$, only delay optimization is done, and when $0 < \mu_e < 1$, $0 < \mu_d < 1$, the tradeoff between the two performance is optimized.

The evaluation function is defined as:

$$f = \mu_e E(\varphi) + \mu_d D(\varphi). \tag{10}$$

The multi-objective programming problem is transformed into $\min\limits_{\varphi \in MAP}(f)$.

## 4. Establishment Process and Mapping Process of TI_GA

### 4.1. Establishment Process of TI_GA

Since the design of NoC's mapping algorithm is an NP-category problem, how to efficiently and quickly obtain near-ideal mapping results without exhausting all schemes is an important aspect to test the performance of the algorithm. Recent research has found that of population intelligence algorithms such as genetic algorithm (GA) [17], particle swarm optimization (PSO) [18], ant colony optimization (ACO) [19], etc. swarm intelligence algorithm has better performance. It is a relatively common genetic algorithm, it mainly simulates the process of evolution and principle of evolution that the most suitable survives in the biological world. In the genetic process, the genes of the better

individuals in the current population are retained, combined or mutated. Poor individuals are likely to be directly eliminated, thus ensuring continuous optimization and continuous improvement of the entire population. The main features include: the use of global search mode, the specific scope is not limited; heuristic search, easy to achieve; there is a certain degree of parallelism. These characteristics make the genetic algorithm in the practical application of NoC mapping solution have great value [20].

However, there are also some problems of genetic algorithms in the process of solving:

1. To enhance the quality of initial population and speed up the convergence by building a superior initial population, we attempt to generate the initial population by utilizing the greedy algorithm.
2. Genetic algorithm can be easily caught in a local optimal solution at the later stage. This phenomenon is ascribed to the obvious decrease of gene types in the population caused by the genetic algorithm, which leads to the decrease of population diversity in the later period. To solve this problem, we decided to use a dual population genetic mechanism: dual population GA has two sub-populations $A$, $B$ which evolve independently according to different mutation and crossover probabilities. For sub-population $A$ crossover probability $P_{c1}$, the value of mutation probability $P_{m1}$ is larger, called detection population, focusing on this problem. In a global search, the algorithm maintains a high ability to search new space; the crossover probability $P_{c2}$ of sub-population $B$ and the mutation probability $P_{m2}$ have a small value, which is called exploiting population, focusing on a local search, and is used to search excellent chromosome genes in the local scope. In the process of evolution, the two species exchange with each other according to certain requirements. Independent evolution of the two species ensures the population diversity and avoids falling into a local optimal solution.

At present, we have suggested a two-population genetic algorithm (TI_GA) to optimize the initial population, and the algorithm flow chart is shown in Figure 3.

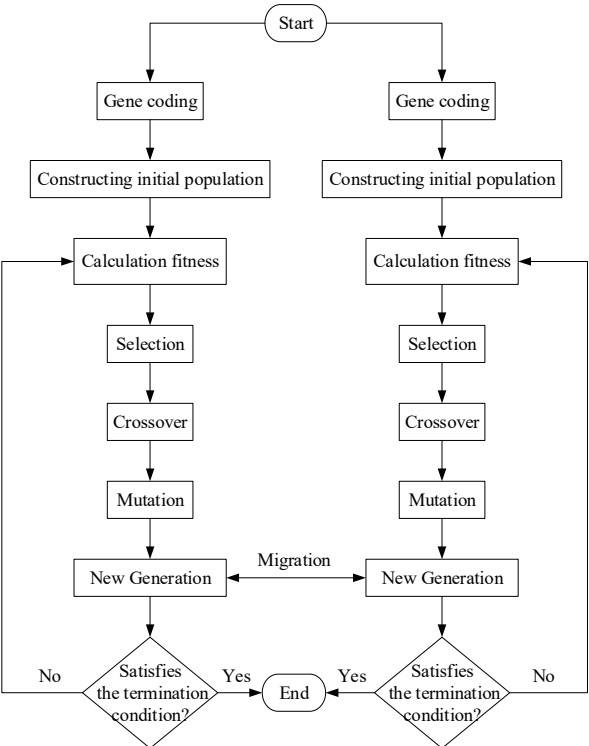

**Figure 3.** Flow chart of two-populations-with-enhanced-initial-population based on genetic algorithm (TI_GA).

### 4.2. Mapping Process of TI_GA

Next, we use TI_GA to realize the mapping process of NoC. It is divided into two steps: the first step is mapping from task to the IP core, the second step is mapping from the IP core to the NoC platform.

### 4.2.1. Mapping Algorithm Steps from Task to IP Core

Step 1. The amounts of tasks and computing units are often different. Assuming that there are N tasks in the current application and M available computing units in the system. Because of the relationship between the amounts of sub-tasks and the amount of computing units, the encoding can be thoroughly discussed under the following three different circumstances.

(a) The amount of computing units is equal to the number of sub-tasks. Chromosome length is equal to the number of subtasks. Suppose that $M = 10$, $N = 10$, chromosome $(5, 4, 2, 1, 6, 8, 10, 9, 7, 3)$ is an available scheduling scheme, in Table 1.

(b) The amount of computing units is larger than the number of sub-tasks. Chromosome length equals the amount of computing units. Suppose $M = 10$, $N = 7$, for example, in Table 2. The IP cores with serial numbers 9, 7 and 3 are not appointed to a task.

(c) The amount of computing units is less than the number of sub-tasks. Suppose $M = 8$, $N = 10$. The coding example is illustrated in Table 3. Task 1 and task 8 are appointed to the IP core with serial number 5, and task 7 and task 10 are appointed to the IP core with serial number 3 [20].

**Table 1.** Encoding example ($M = N$).

| Task ID | 1 | 2 | 3 | 4 | 5 | 6 | 7 | 8 | 9 | 10 |
|---|---|---|---|---|---|---|---|---|---|---|
| **Processing Element ID** | 5 | 4 | 2 | 1 | 6 | 8 | 10 | 9 | 7 | 3 |

**Table 2.** Encoding example ($M > N$).

| Task ID | 1 | 2 | 3 | 4 | 5 | 6 | 7 | × | × | × |
|---|---|---|---|---|---|---|---|---|---|---|
| **Processing Element ID** | 5 | 4 | 2 | 1 | 6 | 8 | 10 | 9 | 7 | 3 |

**Table 3.** Encoding example ($M < N$).

| Task ID | 1 | 2 | 3 | 4 | 5 | 6 | 7 | 8 | 9 | 10 |
|---|---|---|---|---|---|---|---|---|---|---|
| **Processing Element ID** | 5 | 4 | 2 | 1 | 6 | 8 | 3 | 5 | 7 | 3 |

Step 2: A two-population genetic strategy is employed in the algorithm proposed in this paper, so we build two initial populations $A$, $B$, and the probability of the crossover and mutation in the population $A$ is $P_{c1}$, $P_{m1}$, respectively. The probability of the crossover and mutation in the population $B$ is $P_{c2}$, $P_{m2}$, respectively. The major steps of building the initial population are demonstrated as follows:

(a) Generate a number $0 \leq r < n$ at random, where n is defined as the amount of IP cores of the TCG, and allocate r into the first position of the individual X.

(b) $P = \{1, 2, 3, \cdots, n\}$ is an available set. Initialize $P$ and remove r from the available set P.

(c) Traversing the number x in set P, putting x in all the available positions of individual X, calculating X's fitness after putting it in these positions, finding the maximum fitness Fit from these fitness values, recording X in the position j where X's fitness is the largest, and eventually, saving $\langle x, j, Fit \rangle$ in the set F.

(d) Traversing set F, search for the four elements with the biggest Fit value according to Fit, choose an element $\langle x, j, Fit \rangle$ at random from the four elements, put X in the *j*-th position of X, and remove x from set P.

(e) Repeat steps (c) and (d) until emptying set P. At that time, a new individual X is generated and X is added to the population pop.

Step 3. On the foundation of superior initial solution, new chromosomes are generated by successive iterations of selection, crossover and mutation to build a superior population.

(a) The reciprocal of the total energy consumption of the IP core and the system delay was selected as the fitness function. The greater the total energy consumption of the IP core and the system delay, the smaller the fitness, and vice versa. The dominant chromosomes were selected by roulette.

(b) Crossing is a crucial operation to generate new solutions. As shown in Figure 4, two chromosomes are randomly selected and the crossing points are randomly selected. The position of the crossing points of the two chromosomes is the same, which ensures that the length of the chromosome swapping parts are the same. Replacement of genes in the same position (from the beginning of the chromosome to the mutation of the chromosome) between two chromosomes.

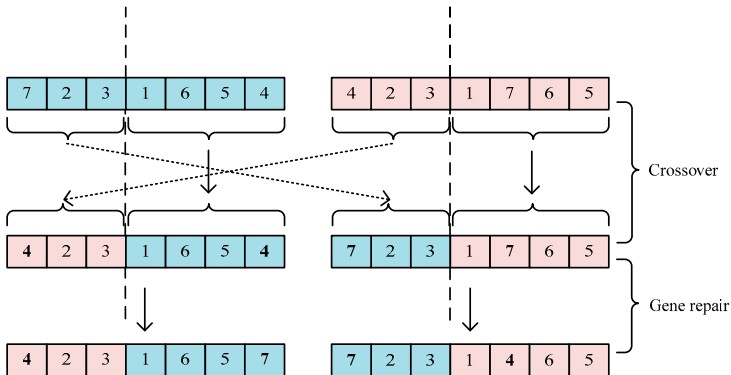

**Figure 4.** Crossover.

(c) The mutation operation refers to the random selection of a chromosome, as shown in Figure 5, and the selection of any gene on that chromosome for mutation. The mutated gene may duplicate the existing gene, which cannot meet the subsequent algorithm conditions. It needs gene repair to ensure that all genes on the chromosome are different.

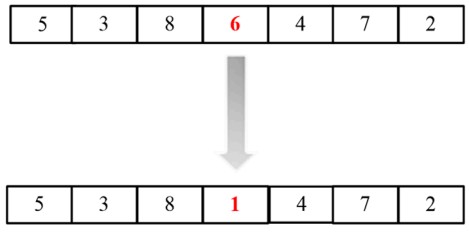

**Figure 5.** Mutation.

Step 4. Dual-population migration: When each generation is finished, it generates a random number *num*, and the optimal chromosomes and *num* chromosomes are selected from *A* and *B* for hybridization, so as to increase population diversity and enhance the global searching capability of the algorithm [20].

### 4.2.2. Mapping Algorithm Steps from IP Core to NoC Platform

To minimize the communication energy consumption and time delay, we implement the location mapping from the IP core to the NoC platform in the second step of the NoC mapping algorithm. We take the set of feasible solutions of the first step as the initial input of the second step of the algorithm. We slightly modify the TI_GA algorithm, and used it in the search process of this step.

In the coding process of the second step mapping, each initial population's individual stands for a chromosome and stands for a mapping solution. Each chromosome is generated by a series of genes, each gene illustrates a node of the NoC structure, and the value of the gene is the amount of processing units selected to map to that node in the set of processing units. In the second step of

mapping, because it is to obtain the mapping scheme from the IP core to the NoC platform aiming at minimizing communication energy consumption and delay, the reciprocal of the result of adding NoC's communication energy consumption and delay is chosen as the fitness function [20].

## 5. Performance Analysis

Our experiment ws carried out on the structure of $4 \times 4$, $5 \times 5$, $6 \times 6$ 2D-Mesh. The routing algorithm adopted the XY routing algorithm. Five examples were used to analyze the scheme, verifying the performance of the proposed mapping scheme.

The first example is MPEG-4 decoding system, the second one is VOPD application, the third one is MWD application, the fourth one is 263enc application, and the fifth one is 263dec application. The attributes of the applied feature graph are shown in Table 4. The mapping was accomplished by TI_GA-based mapping scheme search algorithm. The parameters of the experiment were defined and the corresponding values are displayed in Table 5. We compared the effect of traditional genetic algorithm and ant colony algorithm with TI_GA. In order to obtain more accurate data, we used BookSim [21] simulator. BookSim could obtain our custom traffic, simulate the operation of NoC and obtain energy consumption and delay data. Our experiments were carried out separately from single-objective optimization and multi-objective optimization.

**Table 4.** Attributes of task communication graph.

| TCG | Number of Task Nodes | Number of Communication Edges | Total Communications (MB/s) |
|---|---|---|---|
| MPEG-4 | 12 | 13 | 3466 |
| VOPD | 16 | 21 | 3731 |
| MWD | 12 | 12 | 1120 |
| 263enc | 12 | 12 | 230.214 |
| 263dec | 14 | 19.636 | 19.636 |

**Table 5.** Experimental parameter definition.

| Definition | Value |
|---|---|
| Maximum number of iterations in experiments | 150 |
| Population size | 100 |
| Population A's crossover probability | 0.6 |
| Population A's mutation probability | 0.03 |
| Population B's crossover probability | 0.2 |
| Population B's mutation probability | 0.003 |
| Packet injection rate (flit) | 0.2 |
| Packet size (filt) | 4 |
| Route buffer (filt) | 8 |

Firstly, we carried out the experiment of single-objective optimization. Setting $\mu_e$ to 1 and $\mu_d$ to 0, we caould get the single-objective optimization with power consumption as the goal. As shown in Figure 6, we got the optimal power consumption solutions of the three algorithms when the network scale was $4 \times 4$, $5 \times 5$ and $6 \times 6$, respectively. Among them, the *X*-axis represented five different applications in different network scales, and the *Y*-axis was the power evaluation value from the traditional genetic algorithm. The smaller the value, the better the corresponding performance. Compared to the traditional genetic algorithm, the power consumption of TI_GA in the MPEG-4 application chart decreased by 6.23% and 4.35% compared with the ant colony algorithm on the experimental platform with the network size of $4 \times 4$. In the MWD application chart, the power consumption of TI_GA decreased by 23% compared to the traditional genetic algorithm and 24.14% compared to the ant colony algorithm. In the VOPD application chart, the power consumption of TI_GA decreased by 12.3% compared to the traditional genetic algorithm. Compared to the ant colony

algorithm, the power consumption decreased by 14%. In the 263enc application chart, the power consumption of TI_GA decreased by 16% compared to the traditional genetic algorithm. Compared to the ant colony algorithm, the power consumption decreased by 15.21%. In the 263dec application chart, the power consumption of TI_GA decreased by 12.67% compared to the traditional genetic algorithm, and the power consumption decreased by 13.81% compared to the ant colony algorithm. In the experimental platform with network scale of $5 \times 5$ and $6 \times 6$, in five test cases, TI_GA had different power consumption reductions compared to the traditional genetic algorithm and ant colony algorithm.

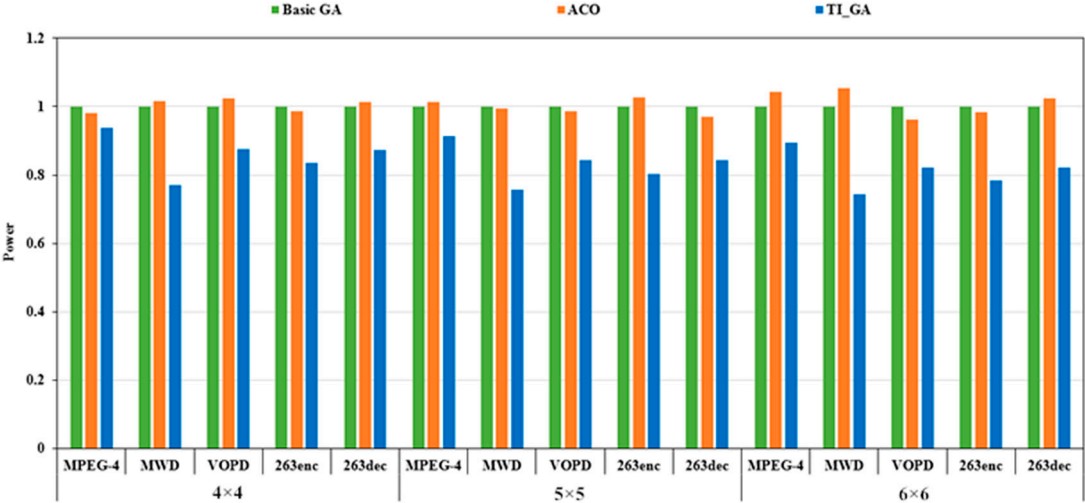

**Figure 6.** TI_GA, traditional genetic algorithm and ant colony optimization (ACO) for MPEG-4, MWD, VOPD, 263enc, 263dec power results comparison ($\mu_e = 1$, $\mu_d = 0$).

Setting $\mu_d$ to 1 and $\mu_e$ to 0, we could get the single-objective optimization with delay as the goal. As shown in Figure 7, we got the optimal delay consumption solutions of the three algorithms when the network scale was $4 \times 4$, $5 \times 5$ and $6 \times 6$, respectively. Among them, the *X*-axis represents five different applications in different network scales, and the *Y*-axis is the delay evaluation value based on the traditional genetic algorithm. The smaller the value, the better the corresponding performance. In the experimental platform with a network size of $4 \times 4$, the delay of TI_GA in the MPEG-4 application graph was 14.2% lower than that of the traditional genetic algorithm. Compared to the ant colony algorithm, the delay decreased by about 8%. In the MWD application chart, the delay of TI_GA was 15.97% lower than that of the traditional genetic algorithm. Compared to the ant colony algorithm, the delay decreased by 18.10%. In the VOPD application chart, TI_GA had a delay of 23.9% compared to the traditional genetic algorithm. Compared to the ant colony algorithm, the delay was decreased by 19.92%. In the 263enc application chart, the delay of TI_GA was 14.55% lower than that of the traditional genetic algorithm. Compared to the ant colony algorithm, the delay was 14.27%. In the 263dec application chart, the delay of TI_GA was 10.95% lower than that of the traditional genetic algorithm. Compared to the ant colony algorithm, the delay was 13.16%. In the experimental platform with network scale of $5 \times 5$ and $6 \times 6$, TI_GA has different degrees of delay in the five test cases compared to the traditional genetic algorithm and ant colony algorithm.



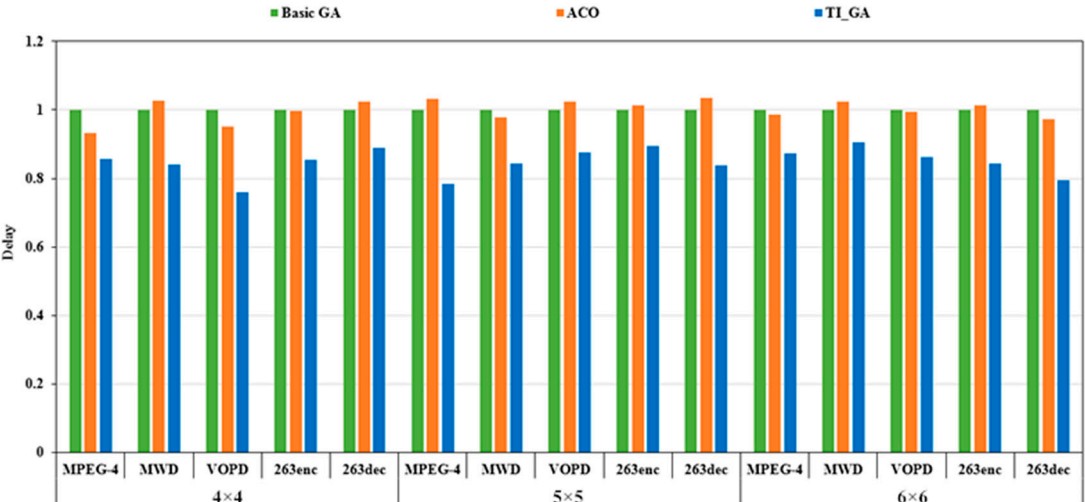

**Figure 7.** TI_GA, traditional genetic algorithm and ACO for MPEG-4, MWD, VOPD, 263enc, 263dec delay results comparison ($\mu_d = 1$, $\mu_e = 0$).

Next, we carried out multi-objective optimization with power consumption and delay as the optimization index under the experimental platform of 4 × 4 network scale. First, we adjusted the values of $\mu_e$ and $\mu_d$ to make $\mu_e : \mu_d = 1 : 1$. This ratio represents a fair tendency to optimize power consumption and latency. Figure 8 shows the optimal power consumption of the three algorithms. The *X*-axis illustrates different applications, and the *Y*-axis is the evaluation from the power consumption achieved by the traditional genetic algorithm. Compared to the traditional genetic algorithm, the power consumption in the MPEG-4 application chart decreased by 3.2%. Compared to the traditional genetic algorithm, the power consumption in the MWD application chart decreased by 13.2%. Compared to the traditional genetic algorithm, the power consumption in the VOPD application chart decreased by 7.6%. As shown in Figure 9, the optimal delay solution obtained by the three algorithms was obtained. The *X*-axis illustrates different applications, and the *Y*-axis is the evaluation from the delay obtained by the traditional genetic algorithm. Compared to the traditional genetic algorithm, the delay in the MPEG-4 application graph decreased by 8.02%. Compared to the traditional genetic algorithm, the delay in the MWD application graph decreased by 7.25%. Compared to the traditional genetic algorithm, the delay in the VOPD application graph decreased by 13.54%.

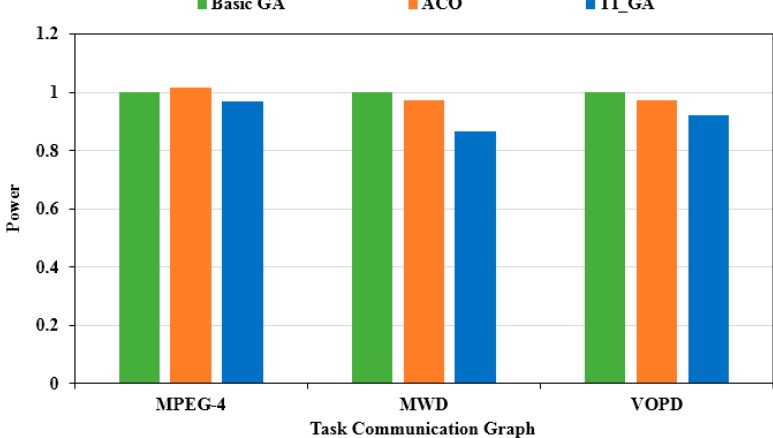

**Figure 8.** TI_GA, traditional genetic algorithm and ACO for MPEG-4, MWD, VOPD power results comparison ($\mu_e : \mu_d = 1:1$).

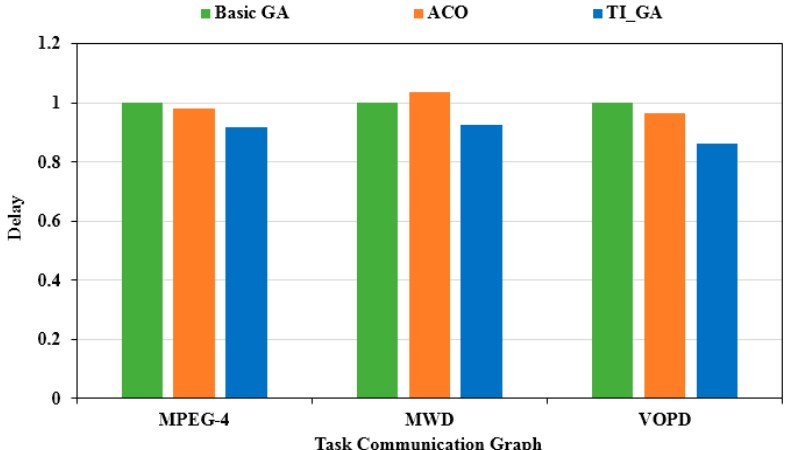

**Figure 9.** TI_GA, traditional genetic algorithm and ACO for MPEG-4, MWD, VOPD delay results comparison ($\mu_e$:$\mu_d$ = 1:1).

We continued to adjust the values of $\mu_e$ and $\mu_d$ to make $\mu_e : \mu_d = 2 : 1$. This ratio represented the direction of optimization for multi-objective optimization, focusing more on power optimization, as shown in Figure 10. The *X*-axis illustrates different applications, and the *Y*-axis is the evaluation based on the power consumption obtained by the traditional genetic algorithm. Compared with the traditional genetic algorithm, the power consumption in the MPEG-4 application chart decreased by 3.75%. Compared with the traditional genetic algorithm, the power consumption in the MWD application chart decreased by 16.15%. Compared with the traditional genetic algorithm, the power consumption in the VOPD application chart decreased by 9.89%. As shown in Figure 11, the optimal solution of the delay obtained by the three algorithms was obtained. The *X*-axis illustrates different applications, and the *Y*-axis is the evaluation based on the delay results of the traditional genetic algorithm. Compared with the traditional genetic algorithm, the delay in the MPEG-4 application graph decreased by 5.49%. Compared to the traditional genetic algorithm, the delay in the MWD application graph decreased by 4.56%. Compared to the traditional genetic algorithm, the delay in the VOPD application graph decreased by 8.23%.

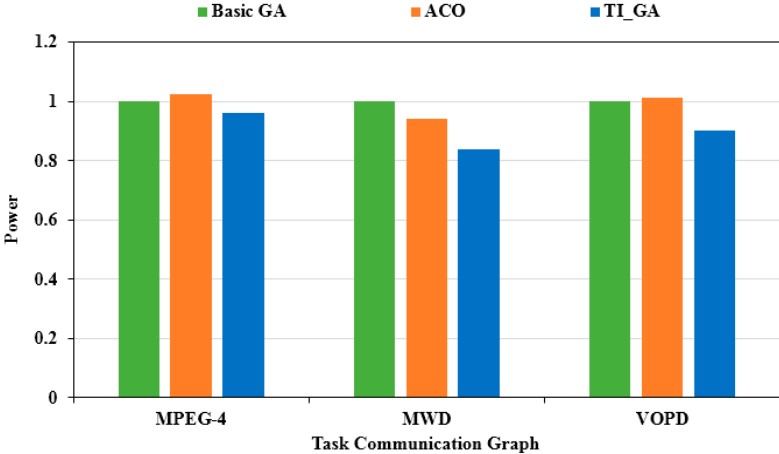

**Figure 10.** TI_GA, traditional genetic algorithm and ACO for MPEG-4, MWD, VOPD power results comparison ($\mu_e$:$\mu_d$ = 2:1).

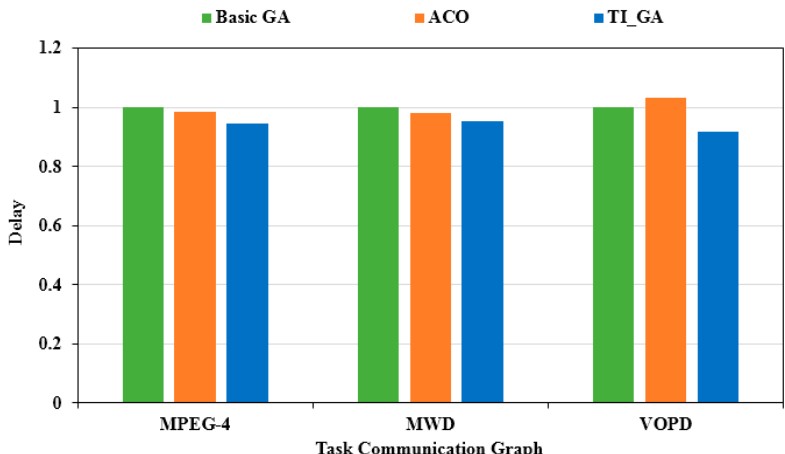

**Figure 11.** TI_GA, traditional genetic algorithm and ACO for MPEG-4, MWD, VOPD delay results comparison ($\mu_e$:$\mu_d$ = 2:1).

We continued to adjust the values of $\mu_e$ and $\mu_d$ to make $\mu_e : \mu_d = 1 : 2$. This ratio represented the multi-objective optimization which focused more on the optimization of delay. As shown in Figure 12, the optimal power consumption of the three algorithms was obtained. The *X*-axis illustrates different applications, and the *Y*-axis is the evaluation based on the power consumption results obtained by the traditional genetic algorithm. Compared to the traditional genetic algorithm, the power consumption in the MPEG-4 application chart decreased by 2.45%. Compared to the traditional genetic algorithm, the power consumption in the MWD application chart decreased by 8.03%. Compared to the traditional genetic algorithm, the power consumption in the VOPD application chart decreasesdby 5.20%. Compared to the case of $\mu_e : \mu_d = 2 : 1$, the optimization of power consumption was not as obvious as the previous group of experiments. As shown in Figure 13, the optimal delay solution by the three algorithms was obtained. The *X*-axis illustrates different applications, and the *Y*-axis is the delay evaluation value based on the traditional genetic algorithm. Compared to the traditional genetic algorithm, the delay in the MPEG-4 application graph decreased by 9.91%; compared to the traditional genetic algorithm, the delay in the MWD application graph decreased by 9.38%; compared to the traditional genetic algorithm, the delay in the VOPD application graph decreased by 18.32%. Compared to the case of $\mu_e : \mu_d = 2 : 1$, the optimization efficiency of delay was improved obviously.

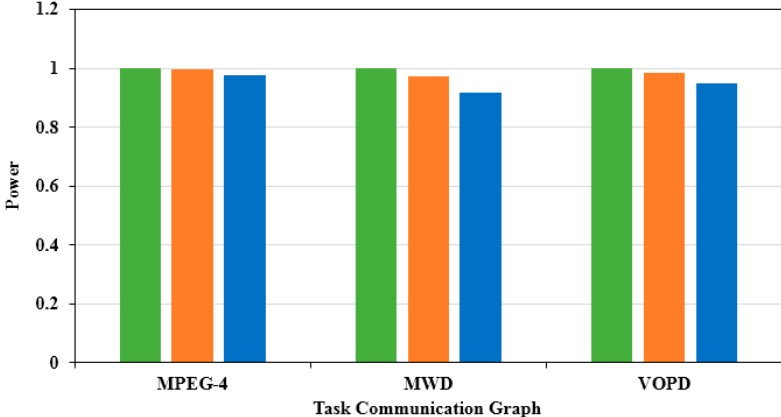

**Figure 12.** TI_GA, traditional genetic algorithm and ACO for MPEG-4, MWD, VOPD power results comparison ($\mu_e$:$\mu_d$ = 1:2).

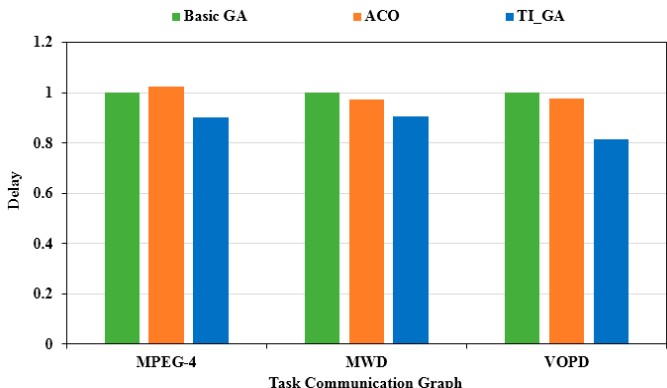

**Figure 13.** TI_GA, traditional genetic algorithm and ACO for MPEG-4, MWD, VOPD delay results comparison ($\mu_e$:$\mu_d$ = 1:2).

Illustrating the influence of $\mu_e$ and $\mu_d$ values on multi-objective optimization, we continued to carry out two groups of experiments on the experimental platform with network scale of $4 \times 4$. As shown in Figure 14, the *X*-axis illustrated the value of $\mu_e$, and the *Y*-axis illustrated the percentage improvement of power optimization performance based on the traditional genetic algorithm. As shown in Figure 15, the *X*-axis illustrates the value of $\mu_d$, and the *Y*-axis illustrates the delay optimization based on the traditional genetic algorithm. With the increase of $\mu_e$ and $\mu_d$, the percentage of power consumption and delay optimization of TI_GA was higher and higher than that of the traditional genetic algorithm. We could weigh the weights between $\mu_e$ and $\mu_d$ according to the needs of applications to maximize power consumption and delay to meet our needs.

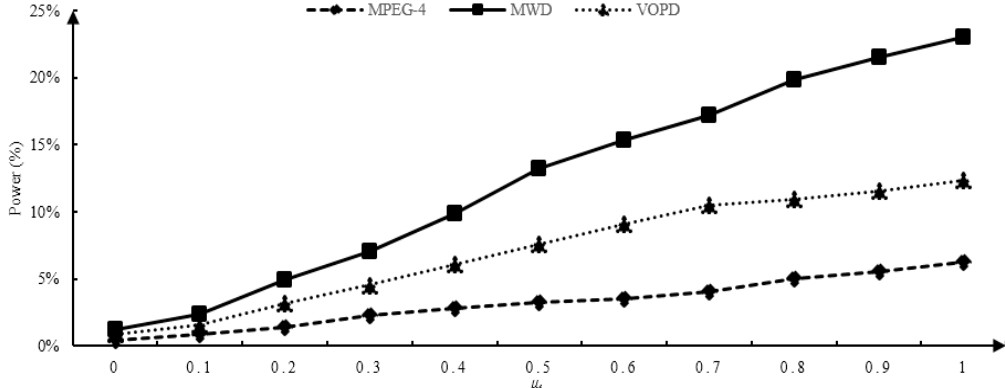

**Figure 14.** Influence of multi-objective optimization parameter $\mu_e$ on power optimization results.

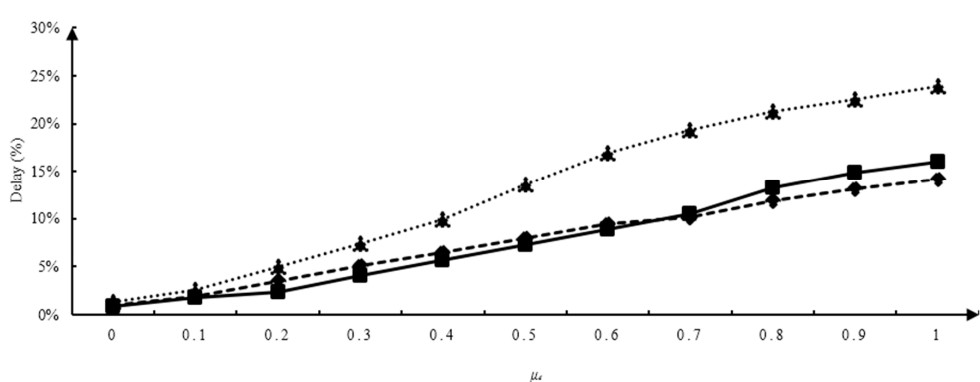

**Figure 15.** Influence of multi-objective optimization parameter $\mu_d$ on delay optimization results.

## 6. Conclusions

NoC also has broad prospects in specific areas of computer science, such as high-performance computing, cloud computing, and artificial intelligence computing, which have corresponding demands for computing performance, power consumption, and other performance optimizations. The development of these areas has also been promoted through the optimization of NoC mapping.

In this paper, we designed the TI_GA optimization algorithm. First, we proposed the power consumption calculation model and system delay model, in view of the two steps NoC mapping. With energy consumption and time delay as optimization objectives, the staged NoC mapping method was used to solve the multi-core heterogeneous cooperative system. In complex multimedia system simulation experiments, the proposed mapping algorithm was superior to the traditional genetic algorithm.

**Author Contributions:** Project administration, J.F.; supervision, J.F.; conceptualization, H.Z. (Huan Zong); methodology, H.Z. (Huan Zong); validation, H.Z. (Huan Zong), H.Z. (Haoyan Zhao); formal analysis, H.Z. (Huan Zong); writing—original draft preparation, H.Z. (Huan Zong), H.Z. (Haoyan Zhao); writing—review and editing, H.Z. (Haoyan Zhao), H.C.

**Funding:** This research was funded by the Beijing Natural Science Foundation, grant number 4192007.

**Acknowledgments:** This work is supported by the Beijing Natural Science Foundation (4192007), and supported by the National Natural Science Foundation of China (61202076), along with other government sponsors. The authors would like to thank the reviewers for their efforts and for providing helpful suggestions that have led to several important improvements in our work. We would also like to thank all teachers and students in our laboratory for helpful discussions.

**Conflicts of Interest:** The authors declare no conflict of interest.

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
