# Peer review of "Intelligent Mapping Method for Power Consumption and Delay Optimization Based on Heterogeneous NoC Platform"

_electronics, doi:10.3390/electronics8080912_

Round 1

Reviewer 1 Report

The paper is interesting and discusses about a scientifically relevant topic. However, I think some important sections or content is missing. Before acceptance, some changes must be applied.

First, I state of the art section is essential. Related works must be identified and discussed, comparing their advantages and problems with the proposed solution. Although some ideas about this point are present in the Introducion, a deeper analysis is needed.

Second, results are very descriptive, but some extensions are needed. In particular, experiments to compare the performance of the proposed solution with statet of the art technologies are required. In the Intriduction it is said SoC are quite unefficcient, Can this be shown in a experiment? That would be very interesting. 

Finally, concluions are too short. Some additional discussions about the impact of the obtained results, future works and other similar questions must be included.

Reviewer 2 Report

The paper presents an optimization algorithm for the Network-on-Chip mapping problem. This problem has large practical relevance. It is NP-complete. The multi-object variant of NoC mapping is considered which has not been studied yet in the literature. The authors apply a modified variant of the Genetic Algorithm to solve the problem. The extensive evaluation shows that the proposed method significantly outperforms the traditional Genetic Algorithm and the Ant Colony Optimization in all cases.

Comments, questions:

- I miss from the model that the finite capacities of the nodes and links are not considered.

- How long did the algorithm run on the presented problem instances?

- Can the algorithm be applied to a larger Mesh as well?

- There are several methods for single-target NoC mapping. I recommend performing experiment to show the benefits of multi-target mapping over single-target mapping.

Further comments:

- I recommend increasing the size of the text labels in figures and charts to make them easier to read.

- Page 4, 3rd paragraph from the bottom, 2nd line: delay on the router -> delay on the routers

Page 7, 1st paragraph, 3rd line: ". the" -> ". The"

- The first sentence of the Section Performance Analysis claims that experiment is carried out on a 4x4 Mesh. You may add 5x5 and 6x6 Meshes as well because these network sizes were also used in some experiments.

Round 2

Reviewer 1 Report

Although I think the content would be much easier to understand considering another organization (including an state of the art section, for example), in its current form the paper may be accepted, as all my concerns have been addressed and, at the end, is up to the authors how the present their contributions.

In conclusion, I think the paper has been highly improved and all my questions answered. The paper may be accepted as it.

Reviewer 2 Report

I accept the answers of the authors.

I recommend the paper for publication.